# Current Advances of Plant-Based Vaccines for Neurodegenerative Diseases

**DOI:** 10.3390/pharmaceutics15020711

**Published:** 2023-02-20

**Authors:** Luis Alberto Bravo-Vázquez, Erick Octavio Mora-Hernández, Alma L. Rodríguez, Padmavati Sahare, Anindya Bandyopadhyay, Asim K. Duttaroy, Sujay Paul

**Affiliations:** 1School of Engineering and Sciences, Campus Querétaro, Tecnologico de Monterrey, Av. Epigmenio González, No. 500 Fracc. San Pablo, Querétaro 76130, Mexico; 2School of Engineering and Sciences, Campus Mexico City, Tecnologico de Monterrey, Calle del Puente, No. 222 Col. Ejidos de Huipulco, Tlalpan, Mexico City 14380, Mexico; 3Instituto de Neurobiología, Universidad Nacional Autónoma de México, Campus UNAM 3001, Juriquilla, Querétaro 76230, Mexico; 4International Rice Research Institute, Manila 4031, Philippines; 5Reliance Industries Ltd., Navi Mumbai 400701, India; 6Department of Nutrition, Institute of Basic Medical Sciences, Faculty of Medicine, University of Oslo, P.O. Box 1046 Blindern, 0317 Oslo, Norway

**Keywords:** neurodegenerative disorders, plant-based vaccines, immunotherapy, Alzheimer’s disease, Parkinson’s disease, molecular farming, novel therapeutic strategies

## Abstract

Neurodegenerative diseases (NDDs) are characterized by the progressive degeneration and/or loss of neurons belonging to the central nervous system, and represent one of the major global health issues. Therefore, a number of immunotherapeutic approaches targeting the non-functional or toxic proteins that induce neurodegeneration in NDDs have been designed in the last decades. In this context, due to unprecedented advances in genetic engineering techniques and molecular farming technology, pioneering plant-based immunogenic antigen expression systems have been developed aiming to offer reliable alternatives to deal with important NDDs, including Alzheimer’s disease, Parkinson’s disease, and multiple sclerosis. Diverse reports have evidenced that plant-made vaccines trigger significant immune responses in model animals, supported by the production of antibodies against the aberrant proteins expressed in the aforementioned NDDs. Moreover, these immunogenic tools have various advantages that make them a viable alternative for preventing and treating NDDs, such as high scalability, no risk of contamination with human pathogens, cold chain free production, and lower production costs. Hence, this article presents an overview of the current progress on plant-manufactured vaccines for NDDs and discusses its future prospects.

## 1. Introduction

Neurodegenerative diseases (NDDs) are a growing cause of mortality and morbidity worldwide, especially in the elderly, making them one of the most significant public health issues [1]. As a matter of fact, it has been projected that at least 70 million people will suffer from NDDs by 2030 and 106 million people will have them by 2050 [2]. NDDs belong to a heterogeneous group of incurable and debilitating human ailments distinguished by the progressive degeneration and/or the death of nerve cells of the central nervous system (CNS), which may result in motor, behavioral, and cognitive deficits [3,4,5,6]. Patients suffering from NDDs tend to experience impaired social functioning, depression, and sleep disorders [7,8,9], while the caregivers of these individuals might face social isolation, stress, burnout, and anxiety [7]. The most representative NDDs are Alzheimer’s disease (AD), Parkinson’s disease (PD), multiple sclerosis (MS), amyotrophic lateral sclerosis (ALS), and Huntington’s disease (HD) [10]. However, frontotemporal dementia (FTD), prion diseases, and dementia with Lewy bodies (DLB) are other types of less common NDDs [11].

Other than aging, there are several agents that affect the onset of both hereditary and sporadic forms of NDDs, ranging from polygenetic to environmental concerns [12,13]. Some of the principal risk factors that have been associated with NDDs are chronic short sleep, sleep disruption [14], stress, lack of exercise, high-fat or high-sugar diets, high cholesterol levels, arterial hypertension, as well as tobacco and alcohol consumption [15]. Remarkably, microwave radiation has been reported to be involved in the manifestation of CNS diseases (e.g., AD) at certain frequencies and exposure times [16]. The pathophysiology of NDDs is characterized by the intracellular or extracellular accumulation of misfolded proteins that lose their biological functions or become toxic, leading to the formation of small oligomeric or large fibrillary aggregates [17,18]. Even so, the molecular mechanisms underlying this conformational transition from healthy and functional proteins into pathological polypeptides are still unknown [19]. Mechanistically, microglial cells and astrocytes (immune cells of the CNS) become active when they recognize these protein aggregates in NDDs. Such activation leads to phagocytosis as well as to the secretion of proinflammatory molecules and reactive oxygen species (ROS) [20,21]. Although inflammation is an immune response that is often beneficial in coping with neuronal injury or pathogen infiltration, chronic inflammatory activity can generate excessive proinflammatory cytokines (e.g., tumor necrosis factor-alpha [TNF-α]) that result in neuronal injury and cell death [22,23]. In the same way, a pro-oxidant environment sustained by chronic microglia activation can greatly affect the progression of neurodegeneration [24]. The complex interplay between abnormal protein aggregates, dysfunctional neurons, and the brain’s resident immune cells results in a vicious cycle of neuronal damage and neuron loss that gradually leads to neurodegeneration (Figure 1).

Despite the fact that several therapeutic strategies have been developed to target the pathophysiological mechanisms of NDDs, these disorders are still incurable, and available treatments only slow down their progression and mitigate their symptoms [25]. Some examples of prospective therapeutic approaches for NDDs include phytochemical-based treatments, non-coding RNA (ncRNA)-centered drugs, mesenchymal stem cell-derived exosomes, histone deacetylases inhibitors, mitochondria-targeted nanoparticles, and molecules targeting the nucleotide-binding oligomerization domain-like receptor family pyrin domain-containing 3 (NLRP3) inflammasome [26,27,28,29,30,31,32].

Remarkably, a number of studies have demonstrated that the immune system or its associated components could be exploited via immunotherapeutic strategies to target the key misfolded and aggregated proteins in NDDs, i.e., beta-amyloid (Aβ), tau, and α-synuclein (α-syn) [33]. In this sense, it has been suggested that antibodies, also called immunoglobulins (Igs), have notable potential as immunotherapeutic agents against NDDs (passive immunization) [34,35,36]. Similarly, peptide-based vaccines employing specific epitopes or peptides that mimic the structure of epitopes (mimotopes) are currently being developed to prevent and/or treat NDDs by harnessing the activity of the immune system (active immunization) [35,36,37]. As an example, in recent years a variety of vaccine designs for NDDs have been described, including MultiTEP-based vaccines, virus-like particle (VLP)-based vaccines, DNA-based vaccines, modified vaccinia virus Ankara (MVA) poxvirus-based vaccines, and synthetic peptide vaccines [38,39,40,41,42,43,44].

Plant-based vaccines are another promising candidate for immunotherapy against NDDs. Plant-derived vaccinations are generally defined as antigenic formulations generated from transgenic plants expressing particular antigens, and are intended to be used as immunogenic systems [45]. Interestingly, plant-based vaccines offer unique advantages that position them as attractive therapeutic tools to prevent and alleviate numerous human diseases. Some of the most important features of these biopharmaceuticals are their low production cost, high safety due to the absence of human pathogens in plants, non-invasive means of immunization via oral vaccines, lack of need for cold chain, extended shelf-life, good scalability, and simple purification processes [46,47,48]. As well, unlike peptides synthesized in eukaryotic expression systems, plant-derived recombinant proteins are often physiologically active due to naturally occurring post-translational modifications in plants [49]. Existing plant-based vaccines against NDDs have been intended mainly to treat AD [50,51]. The first insights in this matter were reported by Kim et al. [52] and Youm et al. [53], who expressed Aβ in transgenic potato (*Solanum tuberosum* L.), demonstrating the prospective use of this technology for AD prevention and treatment. Later, Youm et al. [54] also developed an Aβ expression system in transgenic tomato (*Solanum lycopersicum* L.). Nonetheless, emerging research has shown that these technologies can also be applied in producing plant-made immunogenic antigens for other NDDs such as PD and MS [55].

Therefore, in this manuscript, we portray a general overview of the current panorama of the development of plant-derived immunotherapy for the management of chronic NDDs and discuss some of the most critical points that should be addressed in future investigations in order to drive forward the advancement of plant-based vaccines for these fatal diseases.

## 2. Overview of the Present Landscape of Plant-Based Vaccines

Over the last few years, the production of innovative plant-based vaccines has experienced outstanding growth. Indeed, the plant-made single-chain fragment variable monoclonal antibody (scFv mAb) for hepatitis B virus (HBV), namely PHB-01, which is employed in the production of a recombinant HBV vaccine, was the first licensed for the manufacturing of human vaccinations [56,57]. Since then, numerous plant-based vaccines have been developed to prevent several human diseases, such as human papillomavirus (HPV) infection, gastric cancer, colorectal cancer, breast cancer, influenza, human immunodeficiency virus (HIV) infection, severe acute respiratory syndrome (SARS), and polio, among others [58,59,60].

In 2021, the results of a phase 3 randomized observer-blind trial in healthy adults supported the immunogenic activity and safety of plant-made quadrivalent VLP vaccine against influenza [61]. Meanwhile, in 2022, a phase 1 dose-escalation study demonstrated the safety and immunogenicity of a plant-derived vaccine formulated with a recombinant protective antigen (rPA) associated with *Bacillus anthracis* (i.e., PA83) [62]. These outcomes imply that plant-based vaccines are getting closer to entering the pharmaceutical market. Indeed, the coronavirus disease 2019 (COVID-19) pandemic scenario led to the accelerated generation of pioneering plant-made vaccines capable of dealing with this international health issue. Some of the most representative corporations involved in this pursuit are Medicago Inc., GlaxoSmithKline, iBio Inc., and Kentucky BioProcessing [63]. For example, once the SARS-CoV-2 spike protein sequence became available, Medicago Inc. generated the VLP vaccine candidates in just 20 days [47]. Moreover, phase 1 randomized trials; the phase 2 portion of a phase 2/3 randomized, placebo-controlled studies; and phase 3, multinational, randomized, placebo-controlled trials have indicated that the SARS-CoV-2 spike glycoprotein VLP (CoVLP) expressed in *Nicotiana benthamiana* L. is highly immunogenic, well tolerated, and can effectively prevent COVID-19 [64,65,66].

Furthermore, as a consequence of the emerging interest, in 2020, the worldwide market for plant-based vaccines was estimated to be worth 927 million USD, and it is anticipated to grow at a rate higher than 11.7% over the following six years [67]. Furthermore, it has been projected that the production of recombinant proteins in plant systems could cost less than 50 USD per gram, which is more profitable when compared to the production costs in conventional systems that range from 300 to 10,000 USD per gram [68]. Therefore, the aforementioned facts highlight a significant trend towards the plant-based vaccine industry to make big strides in the next decade, which will likely drive the development, improvement, pre-clinical and clinical testing, and market entry of plant-made vaccines for NDDs, as well as for other highly prevalent human illnesses.

## 3. Mechanism of Action of Peptide-Based Vaccines for Neurodegenerative Diseases

The possible mechanism of action of peptide-based vaccines for NDDs begins when antigen-presenting cells (APCs) use specialized receptors known as pattern recognition receptors (PRRs) to identify foreign antigens that have entered the human body following oral or parental immunization [69]. In the case of NDDs, such antigens consist of full-length disease-associated proteins (e.g., Aβ and α-syn) or highly immunogenic fragments thereof [33]. Afterward, antigens are internalized into APCs via receptor-mediated endocytosis or phagocytosis and are processed by the proteasome so that APCs can express them on their surface as peptide epitopes in the major histocompatibility complex (MHC) class II, forming a peptide-MHC II complex [69,70]. Subsequently, T helper cells (Th cells, also called CD4+ T cells) interact with APCs via specialized receptors that recognize peptide-MHC II complexes and through APCs’ co-stimulatory molecules whose ligands are on CD4+ T cells. These events activate CD4+ T cells, allowing them to interact with B cells and stimulate antibody synthesis [71].

It is worth mentioning that Th cells are subclassified into Th1 and Th2 cells. Th1 cells are responsible for the production of Th1-type cytokines (e.g., interferon-gamma [IFN-γ]), which are involved in proinflammatory responses against intracellular pathogens and autoimmune responses [72]. Contrary to this, Th2 cells are associated with the elimination of extracellular pathogens and synthesize Th2-type cytokines (e.g., interleukin-4 [IL-4], IL-5, IL-10, and IL-13) that trigger anti-inflammatory responses, B cell proliferation, Ig class-switching, and antibody production [72,73]. Based on these assertions, most of the peptide-based vaccine designs for NDDs should solely employ Th2 cell-related adjuvants as well as T and B cell epitopes in order to elicit anti-inflammatory humoral immunity and protective antibodies [74,75]. In turn, this prevents the synthesis of Th1 cytokines, which could promote inflammatory responses and cell-mediated immunity (CMI), thus avoiding neuroinflammation and cell death [74,75].

As a matter of fact, B cells are able to internalize antigens by receptor-mediated endocytosis after the antigen binds to surface-immunoglobulin B cell receptors (BCRs); these antigens are then processed, resulting in peptide-MHC II complexes that are displayed on the cell surface [76]. Active CD4+ T cells interplay with those B cells that share the same peptide-MHC II complex and stimulate them to undergo cell proliferation, Ig class-switching, differentiating into plasma cells that can secrete highly antigen-specific antibodies and produce memory B cells [71,76]. Lastly, the produced antibodies might reach and cross the blood-brain barrier through circulation and selectively bind to aberrant disease-associated proteins present in NDDs [50,77]. Once in the brain, the antibodies might mediate different processes to eliminate these proteins, such as blocking the formation of protein aggregates, disabling existing protein aggregates by changing their conformation, and/or acting as markers for microglial cells to phagocytose aberrant proteins and their aggregates [50,77]. Furthermore, vaccines may be focused on inducing the production of antibodies targeting the enzymes responsible for the synthesis of aberrant proteins in order to block their activity [78]. The probable mechanism of action of peptide-based vaccines for NDDs is depicted in Figure 2.

## 4. Progress on Plant-Based Vaccines for Alzheimer’s Disease

AD is a progressive neurodegenerative disorder described as the most prevalent cause of dementia. Age-associated clinical signs include memory impairment and a gradual decrease in the patient’s cognition, behavior, and function. Regarding epidemiology, more than 50 million people are currently suffering from dementia, and by 2050, this number is expected to triple [79,80]. Although little is known about AD etiology, it is characterized by elevated levels of fibrillar Aβ peptides in extracellular senile plaques and the intracellular presence of aggregated tau filaments, forming neurofibrillary tangles [81]. Aβ is a 40–42 amino acid fragment produced from the amyloid precursor protein (APP) by two sequential proteolytic cleavages with β-secretase (BACE1) and γ-secretase [82]. Its longer fragment (Aβ1-42 or Aβ42) is the main constituent of amyloid plaques and has a great propensity to aggregate [51]. In the past years, several studies have focused on developing plant-based immunological therapies against AD by heterologously producing associated proteins such as Aβ and BACE1 as antigenic molecules, which have shown promising results.

In 2010, Ishii-Katsuno et al. [83] developed an oral vaccine expressing green fluorescent protein (GFP)-conjugated Aβ42 in pepper plant leaves (*Capsicum annuum* L. var. *angulosum*) through *Tobamovirus* infection, and tested it in Tg2576 transgenic mice (a Swedish familial AD model with K670N and M671L mutations in APP) and wild-type (WT) B6 mice. The vaccine was delivered by animal gastric intubation and co-administered with cholera toxin B subunit (CTB) as an oral adjuvant. CTB corresponds to a non-toxic component of the cholera toxin that has a high affinity for the monosialotetrahexosylganglioside (GM1) receptor [84]. Since GM1 is present in a wide range of cells, including macrophages, dendritic cells, B cells, gut epithelial cells, and APCs, CTB can be easily recognized by such cells, mediating its entry into the immune system. As a result, CTB is extensively employed in vaccine design, where it is chemically or genetically fused to antigens of interest in order to enhance immune responses [84,85]. As well, CTB inhibits Th1 cell-mediated responses [83]. After immunization, intracerebral Aβ42 and senile plaques were shown to be significantly reduced, and serum anti-Aβ antibody concentration was higher compared to the controls. Interestingly, regarding the vaccine’s safety, the inflammatory Th1 globulin IgG2a was significantly lowered in the orally immunized animals when compared to the controls treated solely with GFP, decreasing the potential to induce inflammatory responses, which usually occur with subcutaneous injections of the same antigen. Furthermore, no notable side effects were observed during the course of the research, but immune persistence was not assessed. Hence, the authors stated that this vaccine could help to develop safe food immunotherapy protocols to prevent senile plaque formation [83].

Similarly, in another study, a BACE1-centered vaccine was produced in tobacco (*Nicotiana tabacum* L.) through chloroplast transformation without affecting the morphological characteristics of the host plant. Subsequently, BALB/c mice that were gavaged with the transplastomic tobacco extracts displayed an immunological reaction to the BACE1 antigen, confirming memory immune cell activation. However, future research should validate BACE1 purity and processing, as well as vaccine immunization efficacy, duration of immunity, and possible side effects in AD animal models [86].

Furthermore, Vitti et al. [87] designed two candidate vaccines against AD disease through bioinformatic approaches, tested their expression in tobacco plants, and assessed their antigenic properties through immunoassays. In this context, the first 15 amino acids of Aβ (Aβ1-15) and the Aβ4-15 fragment were selected as two potential disease-related epitopes. Engineered *Cucumber mosaic virus* (CMV) with the Aβ sequences efficiently infected and replicated within the tobacco leaves. Moreover, Aβ1-42 and Aβ1-15 polyclonal antiserum in mice was synthesized for subsequent peptide identification. However, when analyzed using western blot and immunoelectron microscopy, both epitopes showed different antiserum recognition; the Aβ1-15 peptide reacted with both Aβ1-42 and Aβ1-15 antiserums, whereas Aβ4-15 was detected just by Aβ1-15 antiserum. Although the authors did not perform animal immunization, they stated that the first N-terminus Aβ1-15 residues are crucial to determine Aβ antigenic characteristics and that this study is a primary approach in pursuing a safe and effective AD vaccine [87].

Since appropriate protein folding and post-translational modifications depend on the endoplasmic reticulum (ER), it is essential to understand the role of this organelle in plant-based expression systems. A previous study analyzed the molecular mechanism and expression profile response to ER stress in rice (*Oryza sativa* L.) that recombinantly accumulated Aβ1-42 [88]. After transformation, it was noticed that ER stress derived from aggregated Aβ inhibited the synthesis of seed storage proteins and triggered an opaque and smaller phenotype. Interestingly, there was an appearance of abnormal processing bodies, which have an important function during the mRNA turnover process and selective translation in seedlings [89]. In addition, several binding proteins (BiPs) and protein disulfide isomerases (PDIs) had higher expression levels, which suggests that these proteins are involved in the structural alterations of the processing bodies. Overall, this research advances the current knowledge on ER stress to develop a potential edible AD vaccine, paving the way for future investigations that could increase recombinant Aβ1-42 yield and improve the grain phenotype. In addition, forthcoming investigations should substantiate the efficacy, safety, and duration of protection of this plant-derived vaccine [88].

Intending to find a simpler delivery system for oral AD plant-derived vaccines in humans, Yoshida et al. [90] expressed GFP-Aβ42 in brown rice through the *Agrobacterium*-mediated method and described that Aβ is mainly accumulated in the rice seed aleurone layer. After oral administration of boiled Aβ rice in C57BL/6J mice, the anti-Aβ antibody titer was observed to increase significantly, concluding that AD in mice can be prevented and treated by long-term oral feeding of boiled Aβ rice without reducing the vaccine’s efficacy. It was also stated that rice facilitates control consumption besides offering economic advantages over other edible vaccines. Notwithstanding this, the study did not report data on the side effects (e.g., neuroinflammatory responses) and the duration of immunity of this vaccine [90]. In another study performed by the same research group [91], GFP-Aβ42 was produced in rice and tested in Tg2576 transgenic mice. Following oral immunization, it was found that the anti-Aβ antibody concentration increased while brain Aβ was reduced compared to organisms subjected to subcutaneous administration. Moreover, IgG1 and IgG2a isotype levels established that the oral vaccine induced a non-inflammatory Th2 response, as well as a decrease in serum Aβ42 titer. The Y-maze test, which allows mice to explore three arms of a maze and is motivated by a rodent’s intrinsic interest to explore previously unknown areas [92], was also used to measure the spatial working memory of vaccinated animals. In this regard, orally immunized mice presented a higher alternation than the controls and subcutaneously injected mice, evidencing a memory improvement. Despite this, information regarding the vaccine’s durability of protection was not stated in this investigation [91].

To continue with the same study line, the abovementioned Aβ rice vaccine was orally co-administered along with CTB in WT B6 mice. Compared to the WT control, anti-Aβ antibody in serum was found to be significantly higher in the immunized animals with few or no side effects. Nevertheless, when CTB was employed as an adjuvant, additional rice seed protein antibodies were detected, indicating that this vaccine may induce food allergies. As a consequence, in order to verify the vaccine’s safety, serum IgG1 and IgG2a isotypes of anti-seed protein antibodies were examined to evaluate possible inflammatory reactions, and it was observed that Aβ fed mice generated a non-inflammatory Th2 response [93]. Nonetheless, the authors stated that the production of rice seed protein antibodies might be suppressed when immunological tolerance is induced by constant rice consumption. In addition, different Aβ epitopes (Aβ1-16, Aβ11-28, Aβ25-35, and Aβ1-42) were compared to identify which ones could be recognized by the antibodies synthesized upon vaccination. Interestingly, all proposed Aβ antigens were detected based on anti-Aβ antibody production, yet duration of immunity was not assessed in this investigation. Finally, it was suggested that future vaccines should be improved by constructing antigens that efficiently recognize Aβ N-terminus regions and that fuse oral adjuvants for clinical applications [93].

Later, Kim et al. [94] linked the fruit-ripening-specific E8 promoter to the human *BACE1* gene, lacking the transmembrane domain sequence, and expressed it in tomato fruit via *Agrobacterium* transformation. The storage stability of tomato-derived BACE1 was then assessed under boiling conditions. Although heat treatment significantly degraded BACE1 in the transformed tomato, additional experiments indicated that the antigenicity of the protein remained stable for over 6 months when stored at room temperature, and retained its activity under acidic conditions, highlighting the advantage of using fruits as an organ for long-term protein accumulation. Moreover, it was stated that tomato could be an excellent heterologous host system for recombinant BACE1 accumulation since it can maintain its stability in the gastrointestinal environment. Future tomato-based expression systems are expected to provide stable, long-lasting, and affordable therapeutic proteins. Nonetheless, forthcoming works should characterize BACE1 purification and evaluate its immunogenicity and safety in model animals [94].

Another study [95] described the accumulation of the Aβ epitope (FRHDSGY), previously developed by McLaurin and collaborators [96], in soybean (*Glycine max* [L.] Merr.) seeds. In principle, the Aβ sequence was inserted into the glycinin A1aB1b subunit, one of the most abundant storage proteins in soybean, to act as a carrier. The research group employed a *G. max* transgenic line which was deficient in seed storage proteins and had significant transformation efficiency. After transformation, it was confirmed that soluble protein levels were comparable to or higher than those in WT controls, indicating that such a line could be used to produce pharmaceutical proteins. Furthermore, A1aB1b-Aβ expression and aggregation induced the development of compartments (closed sections within the cell cytosol) directly derived from the ER. The authors concluded that seed storage proteins can be used as carriers to transport antigenic peptides in soybean, and that ER may function as a potential organelle for their stable accumulation. However, the immunogenicity and safety of the Aβ epitope has yet to be examined [95].

The receptor for advanced glycation end products (RAGE) is also suggested to be a therapeutic target for AD due to its role in regulating Aβ translocation across the blood-brain barrier. In 2016, Romero-Maldonado et al. [97] successfully expressed two recombinant chimeric proteins in *N. tabacum* to propose a new plant-based candidate vaccine against AD. For the first antigenic protein, the RAGE23-54 epitope was fused with the *Escherichia coli* heat-labile enterotoxin B subunit (LTB) as an adjuvant carrier (LTB:RAGE). The LTB subunit is used in peptide-based vaccine designs to trigger strong immune responses due to its ability to stimulate the expression of activation markers of the MHC class II on B cells (i.e., B7, CD40, CD25 and ICAM-1). As well, LTB induces the expression of the activation marker CD86 (B7-2) in APCs, which in turn mediates the co-stimulation of CD4+ T cells [98]. The second protein consisted in combining RAGE23-54 with Aβ42 (Aβ:RAGE). However, both transgenic plants exhibited phenotypic alterations after transformation, including lack of flowering and growth retardation, suggesting that these abnormalities may be closely related to a specific RAGE fragment that induces toxicity. Thus, several alternatives were suggested by this research group to foster the expression of target proteins in future inquiries, such as using viral vector-mediated expression and applying inducible or tissue-specific promoters. Moreover, enhancing antigen cytoplasmic expression is also recommended to avoid ER stress while overcoming RAGE23-54 toxicity. Further studies should focus on assessing the immunogenicity and safety of these RAGE-based antigens in AD animal models to evaluate the reduction of pathogenic Aβ brain accumulation [97].

Th cells’ memory to tetanus (Tet) toxoid is present in nearly all humans; therefore, an epitope derived from the tetanus toxoid could elicit robust T-cell stimulatory properties, which are of interest in increasing the efficacy of vaccine systems. Zeltins and colleagues [99] harnessed this principle to develop plant-derived VLPs in lily (*Lilium* sp.) for the treatment of psoriasis, cat allergy, and AD. They incorporated a Tet-derived epitope (TT) into CMV, which resulted in CMV_TT_ being the carrier for the antigen corresponding to each pathology (i.e., interleukin-17 for psoriasis, Fel-d1 for cat allergy, and Aβ for AD). It was observed that CMV_TT_ boosted T-cell response in human primary T cells. Furthermore, the incorporation of the TT increased immunogenicity even under age-limiting conditions in mice, which is of clinical interest due to the suboptimal immune response of elderly vaccine recipients. In the specific case of the AD vaccine, it has been reported that targeting the N-terminal section of Aβ1-42 (Aβ1-6) with antibodies might be an efficient intervention against the disease. Therefore, Aβ1-6 was coupled with CMV_TT_ (Aβ_1-6_-CMV_TT_) or with VLPs lacking the tetanus epitope (Aβ_1-6_-CMV_WT_). As a result, BALB/c mice vaccinated with Aβ_1-6_-CMV_TT_ exhibited stronger immune responses than those immunized with Aβ_1-6_-CMV_WT_; meanwhile, the Aβ_1-6_-CMV_TT_ vaccine‘s immunogenicity was higher in those mice that were previously immunized against tetanus. Additionally, mice antibodies were proven to be specific by being able to recognize Aβ plaques in human AD brain sections. Accordingly, the CMV_TT_ showed great adaptability as a carrier for different antigens. Notably, for the three diseases studied, the proposed plant-based vaccines triggered a robust and selective immune response, even in senior recipients, which is a major public health challenge. Despite this, in this study, no data related to adverse effects or the duration of protection of the vaccine were described [99].

An important feature of AD that should be considered for antibody recognition in vaccine development is the fact that a proportion of the brain’s Aβ variants are N-terminal truncated forms of the full length Aβ1-40 and Aβ1-42 peptides, therefore reduced versions of epitopes must be used to ensure the correct targeting. The L1 protein is a popular VLP and one of the major capsid proteins of HPV; hence, it was used for plant expression of the Aβ11-28 epitope, which is present in both the full-length and modified species of Aβ1-42. The abovementioned epitope was inserted simultaneously in the h4 helix and in the coil region of the VLP, yielding two chimeric molecules (VLP-L1a and VLP-L1b) [100]. Both vectors were successfully transformed in plants, as reported by Uribe-Campero et al. [101] in *N*. *benthamiana*, and soluble proteins were extracted 7 days post-agroinfiltration. Remarkably, contrary to the non-immunized control groups, antibodies of C57BL/6J mice immunized with both chimeric proteins recognized both the full-length and truncated peptides (Aβ11-42, as well as the N-pyroglutamate-modified amyloid beta peptides pyroGluAβ3-42 and pyroGluAβ11-42). In addition, these antibodies recognized amyloid aggregates in amyloid precursor protein (APP)-transgenic mice as well as in postmortem human AD brain tissues. Moreover, the anti-inflammatory Th2 response-associated antibodies IgG1 and IgG2b were detected in sera from immunized mice, but not IgG2c pro-inflammatory ones. Even so, no data on the vaccine’s duration of protection were provided in this article. Finally, researchers hypothesized that the Aβ11-28 epitope could be easily reached by APCs, thus explaining the robust immunogenic response observed in the treated animals. Overall, this report demonstrates the successful plant-based production of a prospective vaccine that does not require adjuvants to induce robust immunogenic properties, since the VLP itself has adjuvating properties. Indeed, the use of VLP as a carrier for an Aβ epitope showed relevant insights into the design of plant-made vaccines for AD [100].

Furthermore, a vaccine candidate for AD that targets RAGE was proposed by Ortega-Berlanga et al. [102]. RAGE has been reported to have an increased expression in AD brain regions and is a promising therapeutic target since it transports Aβ into the nervous system, favoring the formation of conglomerated plaques that can lead to disease progression. In order to design a plant-made vaccine for AD, this research group chose the microalga *Schizochytrium* sp. and *Agrobacterium* to carry out the expression of the vector pAlgevir-LTB:RAGE. Following ethanol induction and protein extraction, the authors confirmed the peptide expression by measuring its immunoreactivity to both the anti-LTB serum and an anti-RAGE monoclonal antibody. Moreover, the thermal stability of the LTB:RAGE antigen was analyzed, as cold-chain free distribution is important for a vaccine to reach all geographic regions. Results showed that the peptide remains stable at 60 °C for 2 h but loses its integrity at 80 °C. Remarkably, fast algal growth makes it possible to yield 6 mg/L culture in a short time (5–8 days), with a relatively inexpensive production cost. In conclusion, the proposed method for the expression of proteins in the microalgae *Schizochytrium* sp. using the Algevir system to produce LTB:RAGE might be a convenient approach for the generation of a time and cost-efficient vaccine against AD. Notwithstanding this, this plant-made vaccine should be further evaluated to measure its immunogenicity, safety, and duration of protection [102].

In 2019, a plant-based disease-modifying therapy for AD prevention was proposed by Kawarabayashi et al. [103]; such a vaccine leverages a transgenic soybean plant that produces 870 mg/g of transgenic soybean seed storage protein with the Th2 epitope Aβ4-10 (Aβ+), which evades T-cell regulated autoimmunity. The Aβ+ protein was produced by inserting tandem repeats of Aβ4-10 in portions of disordered regions of soybean glycinin A1aB1b cDNA. A plasmid containing the cDNA was then transformed into immature soybean embryos. For the immunization, Aβ+ or control protein Aβ- with CTB was administered through a catheter into the guts of 9-week-old until 22–58 weeks old TgCRND8 mice, which corresponds to an AD animal model that shows spatial learning deterioration and higher levels of Aβ accumulation and plaques in the brain. Interestingly, results from the Morris water maze test (MWM) of spatial learning indicated that Aβ+ treated animals had improved learning when compared to Aβ- treated mice, even at 57 weeks of age. Additionally, a reduced amount of Aβ and a decrease in amyloid plaques were reported in Aβ+ immunized mice. Splenocyte proliferation and cytokine release in response to Aβ+ and suppressed glial-mediated inflammation also suggested a sustained and safe activation of innate and humoral immune responses due to Aβ+ oral immunization. Additionally, no histological meningoencephalitis or bleeding were detected in mouse brains. Altogether, the proposed immunotherapy could aid in avoiding the need for booster injections and demonstrates a preventative effect on learning impairment, as well as on the pathological progress of AD. However, the forthcoming work should examine the duration of immunity of this vaccine [103].

Later, Yoshida and colleagues [104], for the first time, genetically modified the herb *Ruta chalepensis* L., a plant rich in secondary metabolites and known for its medicinal properties, including as an anti-inflammatory, a central nervous system depressant, and an antipyretic. Accordingly, the *Agrobacterium*-mediated transformation method was used to successfully introduce the Aβ42 sequence fused with GFP to *R. chalepensis* with the aim of generating an oral vaccine against AD. In fact, GFP was used for the detection of modified plants and as an enhancer of Aβ concentration since it is a larger molecule than Aβ. To examine the immunogenic properties, fresh leaves were fed to male C57BL/6J mice. Subsequently, researchers evaluated the effect of the transformed plant and its co-administration with CTB. Mice treated with the transgenic plant, with or without CTB, showed an increase in antibodies against Aβ in serum without side effects, demonstrating that this particular system does not require an adjuvant. Interestingly, the bioactive compounds found in medicinal herbs may have synergistic effects with the vaccine, enhancing its therapeutic outcome. Altogether, *R. chalepensis* modified for the co-expression of the Aβ peptide gene and GFP has been proposed as a plant-based vaccine for oral administration against AD, yielding an increased amount of AB antibodies in the serum of treated mice even without the addition of the adjuvant CTB. However, the duration of the protection conferred by this vaccine has yet to be analyzed [104]. The general procedure for developing and testing plant-based vaccines against NDDs is shown in Figure 3.

## 5. Progress on Plant-Based Vaccines for Parkinson’s Disease

PD belongs to the synucleinopathies, an important group of NDDs of multifactorial pathogenesis linked to abnormally misfolded aggregates of α-syn, a pervasive intracellular protein found in both the brain and peripheral nervous systems [105,106]. Other types of synucleinopathies are DLB, PD dementia (PDD), and incidental Lewy body disease (ILBD) [105]. In particular, PD is the world’s second most prevalent neurodegenerative condition, affecting at least 6.1 million people worldwide. The neuropathology of this ailment is distinguished by dopaminergic neuron death in the substantia nigra, neuroinflammation, and by the occurrence of Lewy bodies and Lewy neurites induced by α-syn aggregation [107]. PD patients manifest motor symptoms, including bradykinesia, tremor, postural instability, and rigidity, as well as non-motor symptoms, involving sleep disorders, sensory disorders, cognitive impairment, psychotic symptoms, and mood disturbances [108]. Currently, there are no treatments for patients suffering from PD and other synucleinopathies, and the majority of existing therapies are aimed at providing symptomatic relief in order to improve the patient’s quality of life [109,110]. However, a few studies have demonstrated the potential of plant-based vaccines to slow down the development of disease.

To begin with, the toxin subunit LTB has been reported as an efficient immunogenic carrier with adjuvant properties to genetically fuse unrelated epitopes. Therefore, Arevalo-Villalobos et al. [111] developed a plant-based mechanism which could be applied as a production system of a vaccine against PD that expresses an LTB-based chimeric protein carrying α-syn epitopes (i.e., α-syn85-99, α-syn109-126, and α-syn126-140) in tobacco plants. Remarkably, the plant lines showed no phenotypic alterations, which could be due to the T-DNA being set in a silenced locus. After the immunization scheme with tobacco leaf tissue, the BALB/c mice serum was analyzed via ELISA assays and showed significantly higher levels of anti-syn IgG antibodies when compared to the group treated with WT plant material. In total, the levels of LTB-syn (0.15 μg/g fresh weight) were sufficient to elicit anti-syn humoral responses targeting pathogenic brain proteins, which makes this a promising approach for producing vaccines against PD and additional synucleinopathies. Regardless, important aspects of this vaccine, such as its adverse effects (especially neuroinflammation) and the duration of its immunity, must be thoroughly investigated [111].

Arevalo-Villalobos et al. [112] also developed the LTB-syn system in carrot cells (*Daucus carota* L.), a platform used to produce the first biopharmaceutical product approved for human use, the recombinant enzyme Elelyso for treating type 1 Gaucher’s disease [113]. In this method, transformed carrot seedlings efficiently produced the LTB-syn protein in its pentameric form in a yield of up to 2.3 μg per gram of dry weight. Moreover, thermostability experiments showed that the protein remained viable at temperatures of up to 60 °C for 2 h, with loss of integrity at 80 °C [112]. For the immunogenic prime-boosting scheme, BALB/c mice were orally immunized with freeze-dried callus containing the plant-made LTB-syn. Furthermore, they received a boost injection of ovalbumin with α-syn B cell epitopes α-syn85-99, α-syn109-126, or α-syn126-140 peptides (OVA-syn conjugate). Interestingly, a significantly higher immune response in the treated group was noticed as compared to the one given the WT calli, measured by the amount of serum IgG and intestinal IgA responses against either LTB or OVA-syn. Overall, this plant-based orally administered vaccine enhances humoral immune response against PD and other synucleinopathies without a boosting requirement and the need of cold storage, which is economically practical for its widespread distribution. It is worth mentioning that critical features of this vaccine, such as its safety and persistence of immunity, must be comprehensively explored in the future [112]. 

## 6. Progress on Plant-Based Vaccines for Multiple Sclerosis

The leading nontraumatic disabling disease affecting young adults is MS, an autoimmune neurodegenerative pathology confined to the central nervous system. It is characterized by inflammatory demyelination, axonal transection, and gliosis, which leads to physical disability and cognitive impairment [114,115,116]. Although there has been a steady advancement in the therapeutic approaches for MS, the currently available drugs are still insufficient to address the demands posed by the intricate nature of this disease [117]. Accordingly, plant-derived vaccines could be a feasible alternative to treat this disease safely and effectively, but to date, only one plant-made MS-related epitope design has been reported, which implies that there are still directions to be addressed in this field of study.

In continuation, Arevalo-Villalobos et al. [118] used the algae-based Algevir vector to accomplish the expression of the *ms-2a* gene in tobacco plants by the *Agrobacterium* method. The MS-2A protein is made up of the target peptides BV5S2, BV6S5, and BV13S1, which activate T regulatory cells (Tregs) to mediate autoreactive T cells to ease the inflammatory process associated with MS, along with a picornaviral 2A oligopeptide sequence (LLNFDLLKLAGDVESNPG-P) in between, which mediates ribosome skipping to achieve the expression of multiple antigens. Following ethanol induction driven by the promoter *AlcA*, up to 0.5 μg of MS-2A peptides per gram of fresh leaf tissue were obtained. However, alternative expression approaches should be explored to improve this yield. For in vivo immunogenicity trials, BALB/c mice were primed with fresh leaf tissue and boosted with OVA-TCR-MS conjugate to further induce T cells’ activity. The results showed significantly higher levels of OD in immunized mice compared to the control given the WT plant, suggesting greater antibody induction against the target anti-TCR-MS. Nonetheless, a wider variety of immunogenic trials is needed to explore the effect on Tregs, which will ultimately induce the desired immunological effects. Moreover, the safety and duration of immunity mediated by this vaccine must be explored hereafter. On the whole, the proposed platform is a promising alternative for the expression of multiple antigens, which could advance the development of a vaccine against several diseases, including MS [118]. A general overview of the current status of the development and evaluation of plant-made vaccines against NDDs is presented in Table 1.

## 7. Phytochemical-Based Treatments for Neurodegenerative Diseases

Phytochemicals are structurally diverse plant-derived compounds that have significant nutritional and therapeutic properties. A wide variety of these molecules have been described as bioactive against NDDs, including flavonoids, phenolic derivatives, terpenoids, carotenoids, and alkaloids, among others [119,120]. It has been demonstrated that certain phytochemicals regulate the pathological factors that contribute to AD and PD by reducing oxidative stress, increasing the phagocytic properties of immune cells, raising neurotransmitter concentrations, and preventing neuroinflammation. Furthermore, they exhibit neuroprotection via targeting apoptosis, the accumulation of aberrant proteins, and the disruption of the blood-brain barrier [119,121,122].

Some of the most studied phytochemicals against both AD and PD are epigallocatechin-3-gallate, berberine, resveratrol, quercetin, and limonoids [123]. Especially for AD, luteolin, baicalin, and rutin have been reported to present inhibitory activity against BACE1, whereas myricetin, resveratrol, curcumin, and altenusin can modulate the formation and deposition of neurofibrillary tangles and Aβ plaques [122]. In addition, chrysin, vanillin, asiatic acid, ferulic acid, thymoquinone, ellagic acid, caffeic acid, α-/β-asarone, and theaflavin have been described as promising antioxidant phytochemicals for PD management [124]. Meanwhile, thymoquinone, huperzine A, and rivastigmine (a physostigmine derivative) present bioactivity over MS [125,126]. Furthermore, several phytochemicals play a role in blocking TNF-α expression, such as oxyresveratrol, silibinin, piperine, higenamine, schisandrin A, and gelsemine [127]. Hence, these complex compounds could represent therapeutic alternatives to conventional neuroprotective synthetic drugs.

Interestingly, diverse phytochemicals have been identified to modulate ncRNAs (such as microRNAs, small interfering RNAs, piwi-interacting RNAs, and long non-coding RNAs) which are involved in NDD development, specifically in protein translation and dysregulated signaling pathways [128]. Likewise, it has been documented that phytochemicals can modulate the expression of genes involved in NDDs via influencing epigenetic mechanisms, such as histone modification, DNA methylation, and by targeting transcription factors [120]. Accordingly, plant species producing anti-NDD phytochemicals could be used as vaccine expression systems in order to enhance their neuroprotective effect, not only at the immunological level, but by directly acting over oxidative processes, inflammatory pathways, and gene expression mechanisms.

## 8. Concluding Remarks

The high complexity of the etiology of NDDs and their substantial prevalence in the world’s population has widely promoted the development of innovative therapies for these life-threatening diseases during the last few years. In this regard, a number of studies have indicated that plant-made vaccines are notable candidates to prevent and alleviate NDDs (including AD, PD, and MS) due to their immunogenic activity observed in model animals. Furthermore, the expression of recombinant therapeutic proteins in plant systems is substantiated by several advantages, including high profitability, broad biosafety, and good scalability. Therefore, the information provided here may pave the way for the development of novel plant-derived vaccines that could help in the management of all NDDs. Nevertheless, the development of plant-based vaccines for NDDs is still in its infancy, and additional efforts must be made to ensure the safety and efficacy of this type of immunotherapeutic approach in extensive clinical trials so that they can be approved for their entry into the pharmaceutical scene.

## 9. Future Directions

As portrayed throughout this current work, significant efforts have been made during the past years in the areas of molecular farming and immunology in order to design advanced plant-based vaccines that can greatly contribute to the prevention and treatment of NDDs. Notwithstanding this, many challenges remain before these plant-formulated vaccines can be approved for use against NDDs. In fact, additional research in suitable animal models is required to comprehensively study the pharmacokinetic/pharmacodynamic properties, dosing, safety, efficacy, duration of immunity, immunization schedule, and possible adverse effects of the therapeutic devices reviewed herein so that these plant-based pharmaceuticals can reach the clinical setting [129,130,131]. As well, since the cold chain is a critical factor associated with the distribution of vaccines and other pharmaceuticals [132,133], thermal stability assays should be performed to guarantee the long-term stability of plant-derived vaccines against NDDs.

In addition, designing more efficient production systems of plant-derived antigens for the prevention and management of NDDs in future studies could be beneficial. For instance, transient plant expression techniques have been found to make it possible to obtain high concentrations of recombinant proteins in a short time [134,135,136]. Another benefit of a transient expression relies on the fact that such technology could be exempted from transgenic regulation due to the absence of gene insertion into the plant genome, resulting in greater public acceptance [137]. However, it has also been suggested that other biotechnological approaches such as RNA interference (RNAi), RNA antisense, and clustered regularly interspaced short palindromic repeat (CRISPR)/CRISPR-associated proteins (Cas; CRISPR/Cas) might assist in improving the yield and quality of recombinant proteins produced in plants by reducing the proteolytic activity of plant cell lines and generating plant specimens capable of humanized glycosylation [138]. Similarly, codon optimization, the proper selection of promoters and terminators, the exploitation of cellular compartments with low proteolytic activity (i.e., apoplast, ER, and chloroplasts), the co-expression of chaperones and folding proteins compatible with the foreign protein expressed, and the inhibition of host silencing mechanisms are other critical factors that must be considered to produce recombinant pharmaceuticals in plants [139].

Undoubtedly, one of the major objectives of generating plant-based vaccines for NDDs is to make the plant itself function as the delivery vehicle for the antigens produced by it. Interestingly, the plant cell wall protects plant-made vaccines in the stomach and aids in being released gradually in the intestine after oral administration [59]. Furthermore, freeze-drying the plant material expressing the epitope of interest can be applied to produce capsules for oral vaccination [140]. On the other hand, even though toxin subunits (e.g., CTB and LTB) and toxin derivatives have been utilized in the design of plant-made vaccinations as adjuvants to enhance mucosal immune responses, there is still too much ambiguity about how the cells of the immunized organism respond to these types of biomolecules [141]; accordingly, innovative methodologies are being designed to overcome such hurdles. In this context, Kim and team [142,143] expressed a tetravalent envelope protein domain III (EDIII) antigen (related to dengue virus) fused to the gut microfold (M) cell target ligand peptide Co1 (tEDIII-Co1) in transgenic rice calli. Since M cells are crucial for antigen sampling without degradation, the Co1 ligand facilitated the recognition and uptake of the antigen at issue, thus triggering strong antibody responses in the absence of adjuvants, as well as robust B and T cell responses in mice subjected to oral immunization with tEDIII-Co1 [143]. Therefore, the M cell-specific peptide ligand Co1 may be exploited in the creation of toxin-free plant-based edible vaccines for NDDs.

In the same way, rice seeds are possible candidates for the production of edible vaccines for NDDs owing to the fact that such kernels are resistant to gastric acid digestion, which facilitates the delivery of the antigens to the intestinal immune system [144]. Moreover, seeds would be one of the best organs to carry out the expression of NDD-associated peptides due to their low content of water, nectar, and other contaminants, reducing harvesting and purification costs [145]. Nevertheless, other plant expression systems that have yet to be explored for the production of oral immunotherapies for NDDs should be evaluated. Such is the case of microalgae, which offers great advantages, such as rapid transformation, high growth rates, post-transcriptional modifications, and low-cost culture media [146,147]. Apart from that, certain plants with anti-neurodegenerative properties could be used in the coming years as production systems for oral vaccines against NDDs in order to generate a possible concomitant effect that could promote the healing process. Some examples of such plants are *Ginkgo biloba* L., *Curcuma longa* L., *Vitis vinifera* L., *Salvia officinalis* L., *Camellia sinensis* (L.) Kuntze, and *Coffea* spp. [148].

It is worth emphasizing that most of the reported plant-made vaccines for NDDs are centered on AD, PD, and MS; hence, further investigations are required to produce plant-based antigens for other classes of NDDs, i.e., ALS, HD, FTD, DLB, and prion diseases. Remarkably, Hung et al. [149] expressed huntingtin exon1 (Htt_ex1_, a protein involved in HD pathogenesis) with abnormally long polyglutamine (polyQ) tracts in tobacco and examined the phenotypic impact of the aberrant protein within the plant. Thus, this innovative approach could be harnessed to promote plant-derived vaccines for HD. Meanwhile, Nworji [150] expressed the recombinant mouse prion protein in tobacco, which may serve as a precedent for devising plant-derived vaccines against prion diseases. Other polypeptides implicated in NDDs are tau (linked with AD and FTD) and TAR DNA-binding protein 43 (TDP-43, related to ALS and FTD) [18], but, to the best of our knowledge, epitopes related to these proteins have not yet been produced in plant systems, thus exhibiting a clear area of opportunity for upcoming experiments. Furthermore, multi-epitope plant-produced vaccines could be very useful for inducing greater immune responses against a single NDD or multiple NDDs simultaneously, as illustrated by Trujillo et al. [151], by generating a chimeric protein containing epitopes related to enterotoxigenic *E. coli*, *Salmonella typhimurium*, and *Vibrio parahaemolyticus* in tobacco. In particular, in the case of NDDs, plant-based vaccine designs with multiple epitopes have been conceived by Romero-Maldonado et al. [97], expressing the RAGE:Aβ complex for AD vaccines. Similarly, Arevalo-Villalobos et al. created a synthetic antigen containing α-syn85-99, α-syn109-126, and α-syn126-140 for PD vaccines [111,112], as well as an antigen made of BV5S2, BV6S5, and BV13S1 peptides for MS vaccines [118].

Another point to consider in forthcoming research relies on the fact that, even though the use of monoclonal antibodies as passive immunization against misfolded toxic proteins has been shown to be a suitable strategy for the treatment of NDDs (e.g., AD-related monoclonal antibodies: Aducanumab, Gantenerumab, Solanezumab, and Crenezumab) [152,153,154,155], plant-based expression systems have not yet been exploited for the production of these antibodies. Remarkably, emerging technologies have made it possible to obtain antibody production yields in whole plants of up to 2 g per kg of fresh weight [156], which portends a profitable future for the entry of this type of pharmaceutical product to the market. In this regard, since the US Food and Drug Administration (FDA) assigned the fast-track designation to the cocktail of three tobacco-produced monoclonal antibodies against the Ebola virus (i.e., ZMapp) [157], the use of plant-made antibodies could expand significantly in the coming years, driving further research and development opportunities for the creation of plant-derived antibodies against NDDs.

Lastly, plant-based vaccines against NDDs could be revolutionized by using plant-associated viruses and their derived proteins. In light of this, it has been shown that the amino acid sequence of the potato virus Y (PVY) nuclear inclusion b protein has a significant homology with the N-terminal region of Aβ, which is highly immunogenic. Mice immunized with PYV-infected potato leaves consistently produced antibodies against Aβ [158]. Furthermore, an interesting investigation elucidated that tobacco consumption triggers the synthesis of antibodies against the tobacco mosaic virus (TMV, a viral vector used to produce proteins of clinical interest [159]) in humans. Specifically, it was determined that such antibodies target the TMV coat protein, which has substantial homology with the human protein translocase of the 40 like outer mitochondrial membrane (TOMM40L) [160]. As a result, the cross-reactivity between anti-TMV antibodies and TOMM40L was observed. Since alterations in the TOM complex (made up of TOMM22, TOMM40, TOMM40L and TOMM70) have been linked with AD and PD pathogenesis, TMV has great potential as a therapeutic tool for those NDDs [160,161]. Indeed, TMV might be utilized as a multifunctional agent that can initiate immune responses against AD and PD both through the selected antigen carried by the said vector and through its associated viral particles [162,163]. These possible future directions associated with promoting the development of plant-derived vaccines for NDDs are illustrated in Figure 4.

In conclusion, to fulfill the need for safe and effective vaccinations for NDDs, it is necessary to explore novel platforms for their production, considering that the benefits of plant-based vaccines, including their affordability, high scalability, and biosafety, may facilitate the creation of innovative immunotherapies for NDDs. Nevertheless, additional investigations are required to thoroughly understand the mechanism of action of these pharmaceuticals inside the human body. It is undeniable that the progress in the development and clinical evaluation of plant-made vaccines for the influenza virus and COVID-19 has offered convincing evidence of how these immunogenic tools might become major game changers for human ailments. As well, the approval of recombinant β-glucocerebrosidase produced in carrot cells for the treatment of Gaucher’s disease (a rare disease associated with mutations in the *GBA1* gene that may be a risk factor for NDD onset [164]) by the FDA [113,135] gives us hope that the progress of these immunotherapeutic strategies will definitely affect the management of NDDs in a positive manner. Thus, we believe that the knowledge provided in this review will significantly contribute to advancing research in this area.

## Figures and Tables

**Figure 1 pharmaceutics-15-00711-f001:**
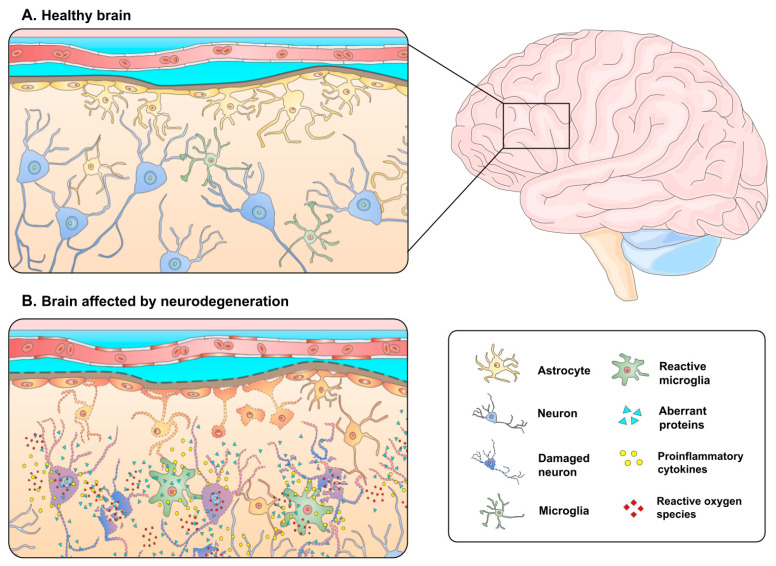
Schematic representation of the pathophysiology of NDDs. (**A**) The microenvironment of a normal brain is characterized by the absence of abnormally expressed proteins and correct nervous function. (**B**) Neurodegeneration is indicated by aberrant protein accumulation, the activation of CNS-resident immune cells, cytokine-mediated neuroinflammation, a pro-oxidative environment, nerve cell damage, and immune-mediated neuronal death.

**Figure 2 pharmaceutics-15-00711-f002:**
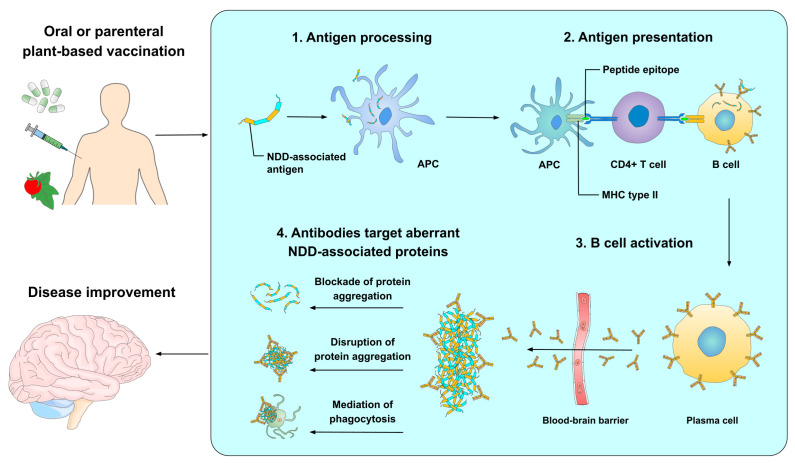
Mechanism of action of peptide-based vaccines against NDDs. Following oral or parental vaccination, APCs use PRRs to identify foreign antigens; these are internalized and processed by the proteasome to be expressed on the cell surface as peptide epitopes in the MHC type II. Afterward, CD4+ T cells interact with APCs, prompting the activation of B cells. As a result, plasma cells (active B cells) produce highly specific antibodies against the NDD-related epitope. Antibodies move through the blood circulation and pass the blood-brain barrier to reach the CNS, where they bind to the NDD-associated proteins in order to eliminate them or impede their functions.

**Figure 3 pharmaceutics-15-00711-f003:**
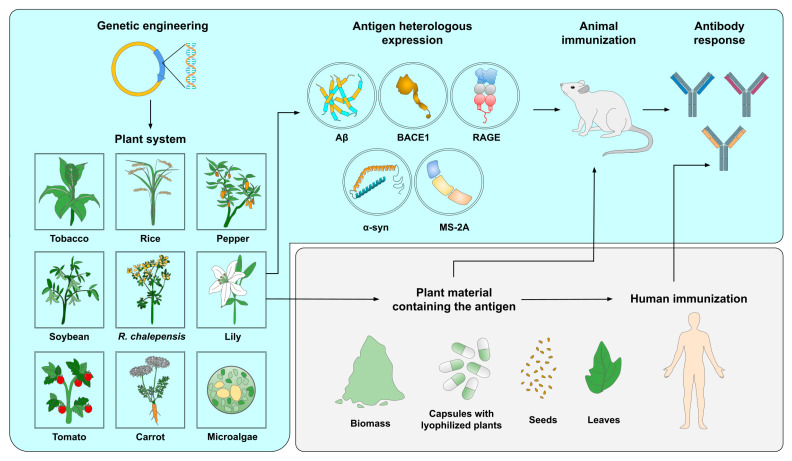
Production of plant-made vaccines against NDDs. Firstly, genetic engineering techniques are employed to generate plant expression systems. Upon heterologous antigen expression, animal models are immunized using either the purified antigens, seeds, plant organs, or lyophilized plants, either as biomass or encapsulated. Alternatively, at the clinical level, the goal is to be able to safely immunize humans with plant-derived vaccines, so this can result in an efficient antibody response for prevention and treatment against NDDs.

**Figure 4 pharmaceutics-15-00711-f004:**
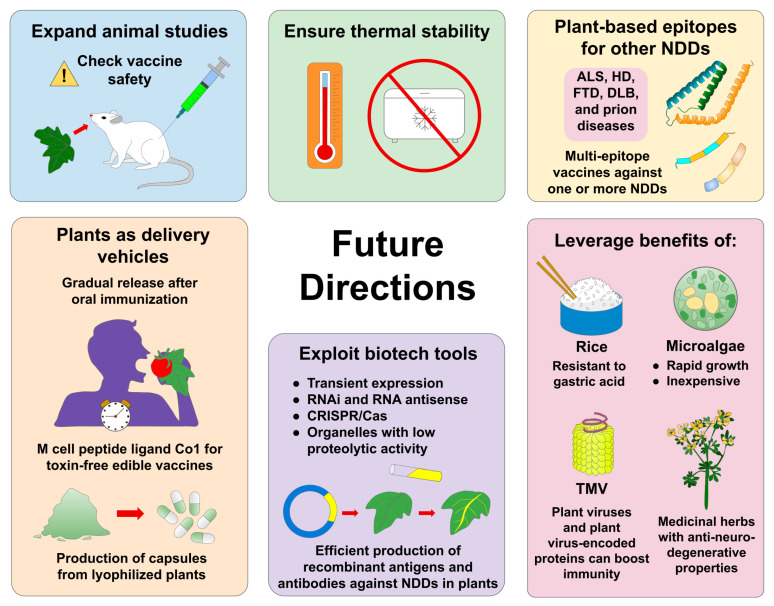
Future directions for the development of plant-based vaccines targeting NDDs. Extensive animal studies are required to ensure plant-derived vaccines’ safety. Furthermore, thermal stability studies should be performed to avoid the need for cold-chain transportation and storage. Additionally, future research should focus on developing plant-made vaccines against other NDDs and exploring the possibility of combining different epitopes in a single vaccine to deal with multiple NDDs concurrently. One of the main advantages of plants is that they can serve as a delivery vehicle for pharmaceuticals, so this property must be taken advantage of and improved to facilitate antigen recognition in the immune cells of the digestive system. On the other hand, several biotechnological techniques could be harnessed with the aim of efficiently producing recombinant NDD-related antigens and antibodies in plants. Finally, upcoming investigations could leverage the advantages of systems such as rice, microalgae, plant viruses, and medicinal plants.

**Table 1 pharmaceutics-15-00711-t001:** Principal findings achieved with plant-based vaccines targeting different NDDs.

Disease	Plant Species	Disease-Related Protein/Epitope Expressed	Biological Model	Main Results	Reference
Alzheimer’s Disease	Pepper (*Capiscum annuum* L. var. *angulosum*)	Aβ42	Tg2576 transgenic and WT B6 mice	Oral immunization prompted the production of anti- Aβ antibodies and the reduction of Aβ levels without inflammatory responses	[83]
	Tobacco (*Nicotiana tabacum* L.)	BACE1	BALB/c mice	Oral immunization triggered a mild production of primary anti-BACE1 antibodies	[86]
		Aβ1-15 and Aβ4-15	-	CMV can be genetically engineered to produce prospective plant-derived vaccines against AD	[87]
	Rice (*Oryza sativa* L.)	Aβ42	-	Expression of recombinant Aβ peptide caused ER stress in rice	[88]
		Aβ42	C57BL/6J mice	Oral immunization increased the serum anti-Aβ antibody titer even in those groups fed with boiled rice	[90]
		Aβ42	Tg2576 transgenic mice	Oral immunization increased the anti-Aβ antibody titer, decreased both intracerebral and serum Aβ levels, and improved mice memory without inflammatory responses	[91]
		Aβ42	WT B6 mice	Oral immunization triggered the production of anti-Aβ antibodies without inflammatory responses	[93]
	Tomato (*Solanum lycopersicum* L.)	BACE1	-	Recombinant BACE1 protein preserves its activity for long periods of storage at cold or room temperature, is stable in low acid conditions, and could be used to produce prospective plant-derived vaccines against AD	[94]
	Soybean (*Glycine max* [L.] Merr.)	Aβ epitope (FRHDSGY)	-	Utilizing seed storage proteins as carriers, the ER may be a promising organelle for the stable accumulation of disease-related peptides	[95]
	Tobacco (*Nicotiana tabacum* L.)	RAGE23-54 and Aβ42	-	Chimeric proteins containing RAGE and/or Aβ epitopes can be produced in tobacco, maintaining their antigenic properties	[97]
	Lily (*Lilium* sp.)	Aβ1-6	C57BL/6 and BALB/c mice	Immunization triggered the production of anti-Aβ antibodies	[99]
	Not specified	Aβ11-28	C57BL/6J mice	Chimeric proteins containing the Aβ epitope triggered the production of anti-Aβ antibodies capable of detecting Aβ plaques in APP-tg mouse and AD brains without inflammatory responses	[100]
	Microalga *Schizochytrium* sp.	RAGE23-54	-	*Schizochytrium* sp. is a reliable platform for the synthesis of thermostable recombinant proteins with an antigenic activity that could be used to produce prospective plant-derived vaccines against AD	[102]
	Soybean (*Glycine max* [L.] Merr.)	Aβ4-10	TgCRND8 mice	Oral immunization triggered the production of anti-Aβ antibodies and prevented spatial learning decline without inflammatory responses	[103]
	*Ruta chalepensis* L.	Aβ42	C57BL/6J mice	Oral immunization triggered the production of anti-Aβ antibodies and since *R*. *chalepensis* is rich in bioactive compounds, it could have synergetic effects as a plant-based vaccine system against AD	[104]
Parkinson’s Disease	Tobacco (*Nicotiana tabacum* L.)	α-syn85-99, α-syn109-126, and α-syn126-140	BALB/c mice	Oral immunization triggered anti-syn humoral responses targeting brain polypeptides, implying the production of antibodies against α-syn; this system could be used to produce plant-made vaccines against PD	[111]
	Carrot (*Daucus carota* L.)	α-syn85-99, α-syn109-126, and α-syn126-140	BALB/c mice	Oral immunization triggered anti-syn humoral responses; this system could be used to produce plant-made vaccines against PD	[112]
Multiple Sclerosis	Tobacco (*Nicotiana tabacum* L.)	MS-2A (containing BV5S2, BV6S5, and BV13S1 peptides)	BALB/c mice	Oral immunization triggered anti-MS-2A humoral responses; this system could be used to produce plant-made vaccines against MS	[118]

## Data Availability

Not applicable.

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
