# Peer review of "Current Advances of Plant-Based Vaccines for Neurodegenerative Diseases"

_pharmaceutics, 2023, doi:10.3390/pharmaceutics15020711_

Round 1

Reviewer 1 Report

Peptide-based vaccines derived from plants may be promising candidates for active immunization against neurodegenerative diseases (NDDs). The main advantages are the low production costs, high safety and non-invasive method of immunization which could therefore make them prime candidates for the treatment of various diseases, not only NDDs, and also in poorer countries.

Focussing on this theme, the authors report a list of studies carried out for the development and efficacy analysis of different plant-based vaccines for AD, synucleinopathies and MS. However, the whole paper lacks essential parts of these reported studies, like side effects, safety, efficacy and duration of immunity, reportable data especially for the phase 1 and phase 3 studies cited.

Furthermore, the inclusion of a more descriptive view of disease pathogenesis is appreciable, particularly with regard to the target molecules of vaccines.

Finally, it would be interesting to report some of the effects of these vaccines also on secondary causes of disease (i.e. variation of neuroinflammation).

Author Response

Peptide-based vaccines derived from plants may be promising candidates for active immunization against neurodegenerative diseases (NDDs). The main advantages are the low production costs, high safety and non-invasive method of immunization which could therefore make them prime candidates for the treatment of various diseases, not only NDDs, and also in poorer countries.

Comment 1: Focussing on this theme, the authors report a list of studies carried out for the development and efficacy analysis of different plant-based vaccines for AD, synucleinopathies and MS. However, the whole paper lacks essential parts of these reported studies, like side effects, safety, efficacy and duration of immunity, reportable data especially for the phase 1 and phase 3 studies cited.

Answer 1: We greatly appreciate the hon'ble reviewer's insightful comments and suggestions very much. It is worth mentioning that many of the studies discussed in this review on NDDs correspond to very early stages of research in which some factors, such as immunogenicity, duration of immunity, or adverse effects caused by vaccination, have not been addressed. Moreover, none of the studies analyzed in this review correspond to clinical trials (1, 2, or 3) since they have all been carried out in mice as preclinical models. Nevertheless, we have highlighted the fact that further analyses are required to ensure the vaccines´ safety and measure the persistence of immunity both in the sections in which we review the original reports and in the section Future Directions of the revised MS (lines 627-631).

Comment 2: Furthermore, the inclusion of a more descriptive view of disease pathogenesis is appreciable, particularly with regard to the target molecules of vaccines.

Answer 2: A more in-depth description of the pathogenesis of NDDs is now provided in the Introduction section of the revised MS (lines 61-74).

Comment 3: Finally, it would be interesting to report some of the effects of these vaccines also on secondary causes of disease (i.e. variation of neuroinflammation).

Answer 3: Due to the fact that this field of study is still in its infancy, very few studies have reported side effects related to neuroinflammation. Nevertheless, as described in the section ´Mechanism of Action of Peptide-Based Vaccines for Neurodegenerative Diseases´ of the revised MS (lines 178-189), peptide-based vaccine designs for NDDs must contain both T and B cell epitopes as well as Th2 cell-related adjuvants in order to prevent inflammatory responses after immunization and promote anti-inflammatory mechanisms. These requirements were considered by diverse authors (see references: 83: 10.1016/j.bbrc.2010.07.120;  91: 10.1016/j.vaccine.2011.06.073; 93: 10.1271/bbb.100861; 100: 10.1007/s10787-017-0408-2; 103: 10.3233/JAD-190023), who reported absence of inflammatory responses in their studies and anti-inflammatory processes mediated by Th2 cells. In the rest of the articles reviewed, no data related to the assessment of neuroinflammatory responses were reported.

Reviewer 2 Report

Authors thoroughly investigated the Current Advances of Plant-Based Vaccines for Neurodegenerative Diseases. The manuscript is correctly written. References are up to the mark.  Although some suggestion could be made it more impact full such as:

1.      Parkinson’s disease should be discussed in review because it is most prevalent neurogenerative disease after Alzheimer’s disease.

2.       Mechanism of action need to be included.

3.       Toxins subunits CTB and LTB should be elaborated.

Authors thoroughly investigated the Current Advances of Plant-Based Vaccines for Neurodegenerative Diseases. The manuscript is correctly written. References are up to the mark.   Although some suggestion could be made it more impact full such as:

1.      Parkinson’s disease should be discussed in review because it is most prevalent neurogenerative disease after Alzheimer’s disease.

2.       Mechanism of action need to be included.

3.       Toxins subunits CTB and LTB should be elaborated.

Author Response

Authors thoroughly investigated the Current Advances of Plant-Based Vaccines for Neurodegenerative Diseases. The manuscript is correctly written. References are up to the mark.  Although some suggestion could be made it more impact full such as:

Comment 1: Parkinson’s disease should be discussed in review because it is most prevalent neurogenerative disease after Alzheimer’s disease.

Answer 1: We are very much thankful to the esteemed reviewer for his/her nice overview, along with useful observations and recommendations. The section that corresponded to synucleinopathies in the original MS is now modified in the revised MS to focus mainly on Parkinson’s disease (lines 492-505). However, it is worth mentioning that the authors of the reports discussed in the said section point out that their plant-made vaccine could be applied in the treatment of other synucleinopathies in addition to Parkinson's disease.

Comment 2: Mechanism of action need to be included.

Answer 2: The general mechanism of action of peptide-based vaccines for NDDs is now explained in the revised MS, together with an illustration of the same (lines 164-214).

Comment 3: Toxins subunits CTB and LTB should be elaborated.

Answer 3: The immunogenic functions of toxin subunits CTB and LTB are now properly explained in the revised MS (CTB: lines 237-244 and LBT: lines 361-365).

Reviewer 3 Report

The manuscript “Current advances of Plant-Based Vaccines for Neurodegenerative diseases” submitted by Luis Alberto Bravo-Vázquez a comprehensive review on  recent developments of plant-derived vaccines, with a special focus on targeting neurodegenerative disorders.  The manuscript is well-written, and the information provided is very relevant for the field. In my opinion, the manuscript is suitable for publication in Pharmaceutics in present form. 

Author Response

The manuscript “Current advances of Plant-Based Vaccines for Neurodegenerative diseases” submitted by Luis Alberto Bravo-Vázquez a comprehensive review on  recent developments of plant-derived vaccines, with a special focus on targeting neurodegenerative disorders.  The manuscript is well-written, and the information provided is very relevant for the field. In my opinion, the manuscript is suitable for publication in Pharmaceutics in present form.

Answer: We greatly appreciate the hon'ble reviewer for his/her positive feedback.

Reviewer 4 Report

Review on Current Advances of Plant-Based Vaccines for Neurodegenerative Diseases

I have completed my review of manuscript Pharmaceutics-2184625, entitled, Current Advances of Plant-Based Vaccines for Neurodegenerative Diseases.”

neurodegenerative diseases (NDDs) are progressive neurodegenerative disorder that causes Alzheimer's disease (AD), dementia, and eventually death. There is currently no efficient treatment available to slow or stop the progression of NDDs, particularly AD.

Progressive neuronal loss in particular brain regions is a hallmark of NDDs. The two most prevalent are AD and Parkinson's disease, and in both of these conditions, a clinical diagnosis is made based on tests that have some limitations in terms of differentiating between similar neurodegenerative disorders and spotting early signs of the disease. Finding new techniques that enable the treatment of such diseases is therefore crucial. In this context, the authors provided a literature review of the current progress on plant-manufactured vaccines for NDDs and discusses its prospects.

In my opinion, the review is important and timely to address some important factors against NDDs. Overall, I am positive about this manuscript but before making a positive decision, I have some concerns and comments about the present form of the manuscript that must be addressed first.

Comments for authors

Comment 1: NDDs can be caused by a variety of factors. When comparing the number of factors that cause NDDs, the authors' description offered in the background (introduction) is insufficient to convey the information, which should be increased for new readers in a review article. The microwave was also thought to be responsible for NDDs, especially AD. I encourage authors to add some background on this topic. The suggested article may assist authors in expanding their background knowledge and understanding the mechanisms by which the EM field interacts with and affects biological systems for various effects.

Article: Microwave Radiation and the Brain: Mechanisms, Current Status, and Future Prospects. International Journal of Molecular Sciences vol. 23 (2022). [https://doi.org/10.3390/ijms23169288].

Comment 2: It was mentioned in the overview of Plant-based vaccines line number 108 that Plant-based monoclonal antibodies as plant vaccines to cure HBV, HPV, and HIV diseases. it is important to define plant-based monoclonal antibody procedures which are a family of medicinal plants ongoing in genetic engineering research to combat these serious NDDs diseases.

Comment 3: This review is based on plant vaccines; authors should increase the explanation of the phytochemical-based treatment to cure the NDDs.

Comment 4: This review concentrated on NDDs, but only one disease, AD, was fully explained. Increased the amount of text on other neural diseases like PD, ALS, etc. Additionally, future considerations added a little more information about the use of plant vaccines as a treatment for neural diseases.

Comment 5: The paper contains errors and typos that make it difficult to understand and distort its intended meaning. I encourage authors to reread carefully and fix any grammatical errors.

Author Response

Review on Current Advances of Plant-Based Vaccines for Neurodegenerative Diseases. I have completed my review of manuscript Pharmaceutics-2184625, entitled, “Current Advances of Plant-Based Vaccines for Neurodegenerative Diseases.”

Neurodegenerative diseases (NDDs) are progressive neurodegenerative disorder that causes Alzheimer's disease (AD), dementia, and eventually death. There is currently no efficient treatment available to slow or stop the progression of NDDs, particularly AD.

Progressive neuronal loss in particular brain regions is a hallmark of NDDs. The two most prevalent are AD and Parkinson's disease, and in both of these conditions, a clinical diagnosis is made based on tests that have some limitations in terms of differentiating between similar neurodegenerative disorders and spotting early signs of the disease. Finding new techniques that enable the treatment of such diseases is therefore crucial. In this context, the authors provided a literature review of the current progress on plant-manufactured vaccines for NDDs and discusses its prospects.

In my opinion, the review is important and timely to address some important factors against NDDs. Overall, I am positive about this manuscript but before making a positive decision, I have some concerns and comments about the present form of the manuscript that must be addressed first.

Comment 1: NDDs can be caused by a variety of factors. When comparing the number of factors that cause NDDs, the authors' description offered in the background (introduction) is insufficient to convey the information, which should be increased for new readers in a review article. The microwave was also thought to be responsible for NDDs, especially AD. I encourage authors to add some background on this topic. The suggested article may assist authors in expanding their background knowledge and understanding the mechanisms by which the EM field interacts with and affects biological systems for various effects.

Article: Microwave Radiation and the Brain: Mechanisms, Current Status, and Future Prospects. International Journal of Molecular Sciences vol. 23 (2022). [https://doi.org/10.3390/ijms23169288].

Answer 1: We highly appreciate the esteemed reviewer's valuable comments and suggestions. A more extensive background specifying those factors that might promote the initiation of NDDs, including the recommended reference (i.e., 10.3390/ijms23169288), is now provided in the Introduction of the revised MS (lines 52-58).

Comment 2: It was mentioned in the overview of Plant-based vaccines line number 108 that Plant-based monoclonal antibodies as plant vaccines to cure HBV, HPV, and HIV diseases. it is important to define plant-based monoclonal antibody procedures which are a family of medicinal plants ongoing in genetic engineering research to combat these serious NDDs diseases.

Answer 2: Although there are promising advances in the development of monoclonal antibodies for NDDs, to date, these antibodies have not yet been produced in plant-based expression systems, representing a significant area of ​​opportunity for further research. This fact is now mentioned in the section Future Directions of the revised MS (lines 707-719).

Comment 3: This review is based on plant vaccines; authors should increase the explanation of the phytochemical-based treatment to cure the NDDs.

Answer 3: An overview of phytochemical-based treatments for NDDs is now provided in the revised MS (lines 575-607).

Comment 4: This review concentrated on NDDs, but only one disease, AD, was fully explained. Increased the amount of text on other neural diseases like PD, ALS, etc. Additionally, future considerations added a little more information about the use of plant vaccines as a treatment for neural diseases.

Answer 4: The section corresponding to synucleinopathies is now rewritten in the revised MS to focus mostly on Parkinson's disease (lines 492-505). It is worth noting that the authors of those works state that their designs could be used to treat various synucleinopathies in addition to Parkinson's disease. On the other hand, it is not possible to include more information concerning plant-based vaccines against other NDDs (e.g., amyotrophic lateral sclerosis, Huntington’s disease, and frontotemporal dementia) since, to date, no such reports have been published. Therefore, we have highlighted this area of ​​opportunity for upcoming studies in the section Future Directions of the original MS (lines 685-697).

Comment 5: The paper contains errors and typos that make it difficult to understand and distort its intended meaning. I encourage authors to reread carefully and fix any grammatical errors.

Answer 5: The manuscript is now thoroughly revised, and linguistic corrections have been made where necessary.

Round 2

Reviewer 1 Report

All comments made were justified.

Reviewer 4 Report

The authors have addressed all of my comments and concerns. I recommend accepting the paper in its present form